# Interlaboratory comparison for the Filovirus Animal Nonclinical Group (FANG) anti-Ebola virus glycoprotein immunoglobulin G enzyme-linked immunosorbent assay

**Michael S. Anderson**[1]*, **Nancy A. Niemuth**[1], **Carol L. Sabourin**[1¤], **Christopher S. Badorrek**[2], **Callie E. Bounds**[2], **Thomas L. Rudge, Jr.**[1]

**1** Battelle, West Jefferson, Ohio, United States of America, **2** U.S. Department of Defense (DOD) Joint Program Executive Office for Chemical, Biological, Radiological, and Nuclear Defense (JPEO-CBRND), Joint Program Manager for Chemical Biological Radiological and Nuclear Medical (JPM-CBRN Medical), Fort Detrick, Maryland, United States of America

¤ Current address: Biomedical Advanced Research and Development Authority (BARDA), Office of the Assistant Secretary for Preparedness and Response (ASPR), U.S. Department of Health and Human Services (HHS), Washington, DC, United States of America

* andersonms@battelle.org

## Abstract

The need for an efficacious vaccine against highly pathogenic filoviruses was reinforced by the devastating 2014–2016 outbreak of Ebola virus (EBOV) disease (EVD) in Guinea, Sierra Leone, and Liberia that resulted in over 28,000 cases and over 11,300 deaths. In addition, the 2018–2020 outbreak in the Democratic Republic of the Congo currently has over 3,400 cases and over 2,200 deaths. A fully licensed vaccine and at least one other investigational vaccine are being deployed to combat this EVD outbreak. To support vaccine development and pre-clinical/clinical testing a Filovirus Animal Nonclinical Group (FANG) human anti-EBOV GP IgG ELISA was developed to measure anti-EBOV GP IgG antibodies. This ELISA is currently being used in multiple laboratories. Reported here is a characterization of an interlaboratory statistical analysis of the human anti-EBOV GP IgG ELISA as part of a collaborative study between five participating laboratories. Each laboratory used similar method protocols and reagents to measure anti-EBOV GP IgG levels in human serum samples from a proficiency panel consisting of ten serum samples created by the differential dilution of a serum sample positive for anti-GP IgG antibodies (BMIZAIRE105) with negative serum (BMI529). The total assay variability (inter- and intra-assay variability) %CVs observed at each laboratory ranged from 12.2 to 30.6. Intermediate precision (inter-assay variability) for the laboratory runs ranged from 8.9 to 21.7%CV and repeatability (intra-assay variability) %CVs ranged from 7.2 to 23.7. The estimated slope for the relationship between $\log_{10}$(Target Concentration) and the $\log_{10}$(Observed Concentration) across all five laboratories was 0.95 with a 90% confidence interval of (0.93, 0.97). Equivalence test results showed that the 90% confidence interval for the ratios for the sample-specific mean concentrations at the five individual labs to the overall laboratory consensus value were within the equivalence bounds of 0.80 to 1.25 for each laboratory and test sample, except for six test

**Data Availability Statement:** All relevant data are within the manuscript and its Supporting Information files.

**Funding:** This project has been funded in whole or in part with funds from the U.S. Department of Defense (DOD) Joint Program Executive Office for Chemical, Biological, Radiological, and Nuclear Defense (JPEO-CBRND) Chemical Biological Radiological and Nuclear Medical (JPM-CBRN Medical) under Battelle contract number GS00Q140ADU402, delivery order W911QY-16-F0074. The funding body contributed to study design and in the decision to submit the article for publication. The views expressed here are those of the authors and do not necessarily represent the views or official position of the DOD.

**Competing interests:** All authors listed on the manuscript have read the manuscript and approve its submission to PLOS ONE. We have no issues relevant to your journal policies for publication, and the authors have no potential competing interests to declare. Our (MSA, NAN, TLR) employment by Battelle, a non-profit contract research organization, does not alter our adherence to PLOS ONE policies on sharing data and materials nor played a role in the study.

samples from Lab D, two samples from Lab B1, and one sample from Lab B2. The mean laboratory concentrations for Lab D were less than those from the other laboratories by 20% on average across the serum samples. The evaluation of the proficiency panel at these laboratories provides a limited assessment of assay precision (intermediate precision, repeatability, and total assay variability), dilutional linearity, and accuracy. This evaluation suggests that the within-laboratory performance of the anti-EBOV GP IgG ELISA as implemented at the five laboratories is consistent with the intended use of the assay based on the acceptance criteria used by laboratories that have validated the assay. However, the assessment of between-laboratory performance revealed lower observed concentrations at Lab D and greater variability in assay results at Lab B1 relative to other laboratories.

## Introduction

The filoviruses (family *Filoviridae*) from the genera *Ebolavirus* and *Marburgvirus* are etiologic agents of sporadic viral hemorrhagic fever outbreaks in humans with high mortality rates. An unprecedented outbreak of Ebola virus (EBOV; species *Zaire ebolavirus*) disease that began in Guinea during December 2013 [1] subsequently spread into neighboring West African countries of Sierra Leone and Liberia, prompting the World Health Organization (WHO) to declare the epidemic a public health emergency of international concern (http://www.who.int/mediacentre/news/statements/2014/ebola-20140808/en/). Phylogenetic analysis of viral isolates from this epidemic suggests a single transmission event introduced the virus, named the EBOV Makona variant [2], from an undetermined natural reservoir into humans in Guinea, followed by transmission between humans to spread the virus throughout Guinea and into Sierra Leone and Liberia [3]. Implementation of containment measures such as patient isolation and improved burial practices eventually controlled the epidemic, which resulted in 28,616 reported cases with a mortality rate of approximately 40% (http://www.who.int/csr/disease/ebola/en/).

The severity of this epidemic and principle transmission from human to human underscored the need for efficacious vaccines (and therapeutics) against EBOV, accelerating the placement of candidate EBOV vaccines into clinical safety trials [4–6]. This need for safe and efficacious vaccines was again evident with the onset of the 10th and largest outbreak in the Democratic Republic of the Congo (DRC) from 2018–2020. The 11th outbreak of EVD continues in the Western DRC.

The characteristics of filovirus infection, where infected patients are contagious only after manifestation of symptoms, allows one to use a ring vaccination strategy for disease containment. Ring vaccination strategy relies on the combination of contact tracing for case identification and a rapid effective vaccine for use in contacts and contacts of contacts of infected patients. The application of this strategy led to the approval of rVSV-ZEBOV (ERVEBO®), a single dose vaccine, using the safety and efficacy data from the clinical trial during the 2014 outbreak in West Africa by the Food and Drug Administration in December 2019 (https://www.fda.gov/news-events/press-announcements/first-fda-approved-vaccine-prevention-ebola-virus-disease-marking-critical-milestone-public-health). The effectiveness of ERVEBO in a ring vaccination response provides an important countermeasure for public health but does not address all unresolved questions in filovirus vaccine utilization including duration of protection, alternate dosing regimens, and the effectiveness of filovirus vaccines based on other viral platforms or alternative strategies. The development of multiple countermeasures

against a disease necessitates the use of a common assay based on a surrogate of protection which can be used to compare the elicited immune response between vaccines and provide valuable information as to the effectiveness and durability of protection. Ideally, this assay is not only informative but simple, reproducible, species independent, and transferrable between labs. For example, during the development of countermeasures against anthrax, a lethal toxin neutralization assay was developed and used by many laboratories [7].

The development of vaccine candidates for Ebola virus disease prophylaxis [8] continues today, including deployment of a heterologous prime boost vaccine with European Commission Market Authorization during the last outbreak. However, the demonstration of efficacy for new filovirus vaccines will be complicated in the absence of a large outbreak and may require evaluation under the FDA Animal Rule or via non-inferiority trials against ERVEBO. Regulatory evaluation using these approaches is only possible with a correlate of protection and a well-developed assay that can measure the response in well-characterized animal challenge models as well as in human clinical trials. The species-neutral ELISA is ideal for bridging data between humans and animal models. Also, since the assay likely will be utilized in multiple experiments at many sites, it is important to demonstrate that the assay is reproducible among different laboratories.

In order to facilitate the development of additional vaccine countermeasures and to address such questions as the durability of immunity, the FANG has supported the development of a human anti-EBOV GP IgG ELISA. This study describes the FANG efforts to determine if the performance of the human anti-EBOV GP IgG ELISA [9] is acceptable for sample evaluation across five participating laboratories. Each laboratory used an anti-EBOV GP IgG ELISA to measure levels of binding in human serum samples from a FANG designed human proficiency panel. The panel consisted of ten human serum samples created by the differential dilution of human serum lot number BMIZAIRE105 (pool of serum with an approximate anti-GP IgG concentration of 1,000 ELISA units/mL) with control human serum (BMI529) without antibody activity. The concentration of the proficiency panel samples ranged from 0.00 ELISA units/mL to approximately 800 ELISA units/mL.

Each participating laboratory received sufficient volume of the proficiency panel for initial testing plus repeats and used their own anti-EBOV GP IgG ELISA established assay. The assay was validated at some laboratories and qualified at others [9]. Data from the participating laboratories were compared by statistical analysis. Both intra-laboratory and inter-laboratory analyses were performed to evaluate repeatability, intermediate precision, dilutional linearity, and accuracy. This paper summarizes both the intra- and inter-laboratory analysis of the results generated in the five separate laboratories. Results from the laboratories are de-identified in the analysis and reported as Laboratory A through E. The repeatability estimate for Laboratory B was greater than the acceptance criteria as established in laboratories that validated the anti-EBOV GP IgG ELISA with human serum, and, as a result, the proficiency panel assay runs were repeated. Results from both the original and repeated runs were included in the analysis and labeled as being from Laboratory B1 and B2, respectively.

## Assay method

A common assay method [9] was tech-transferred to the participating laboratories, but there were minor variations in equipment/materials/procedures between laboratories. The analysis of the proficiency panel in the ELISA was performed similarly at Labs A, B1, and B2. All three used two separate operators on separate days. Samples were analyzed using a starting dilution of 1:62.5 and followed the plate layout as illustrated in Table 1. These plate layouts represent 15

**Table 1. Plate layout used at Laboratories A, B1, and B2.**

| Sample ID | Plate Number | | | | | | | |
|---|---|---|---|---|---|---|---|---|
| | 1 | 2 | 3 | 4 | 5 | 6 | 7 | 8 |
| BMI-ZPP-11 | X (3) | X | | | | | X | X |
| BMI-ZPP-12 | X | X | X | | X | | X (3) | |
| BMI-ZPP-13 | | X (3) | | X | X | X | X | X |
| BMI-ZPP-14 | X | X | X | X | X | X | | X (3) |
| BMI-ZPP-15 | | X | X (3) | | | X | X | |
| BMI-ZPP-16 | | X | X | X | X | X | | X |
| BMI-ZPP-17 | | | X | X (3) | | X | X | X |
| BMI-ZPP-18 | X | | X | X | | X | X | |
| BMI-ZPP-19 | X | | | | X (3) | X | | X |
| BMI-ZPP-20 | X | | | X | X | X | | |

| Sample ID | Plate Number | | | | | | | |
|---|---|---|---|---|---|---|---|---|
| | 9 | 10 | 11 | 12 | 13 | 14 | 15 | |
| BMI-ZPP-11 | X | X | | X | X | X | X | |
| BMI-ZPP-12 | | | X | X | X | X | X | |
| BMI-ZPP-13 | | | X | X | | X | X | |
| BMI-ZPP-14 | | | X | X | | X | | |
| BMI-ZPP-15 | X | X | X | | X | X | X | |
| BMI-ZPP-16 | X (3) | X | | X | X | | | |
| BMI-ZPP-17 | X | X | | X | X | | X | |
| BMI-ZPP-18 | | X (3) | | X | X | X | X | |
| BMI-ZPP-19 | X | X | X | | X | X | X | |
| BMI-ZPP-20 | X | | X (3) | X | X | X | X | |

An "X" indicates that sample was analyzed on the indicated plate.

An "X (3)" (shaded) indicates that sample was analyzed on the indicated plate three times.

plates with specific proficiency panel samples on each plate. All 15 plates were run twice for a total of 30 plates for each of Labs A, B1, and B2.

The analysis of the proficiency panel in the ELISA was performed at Lab C by two separate operators over three days and at Lab D by two separate operators over five days. Samples were analyzed using a starting dilution of 1:50 and followed the plate layout as illustrated in Table 2. These plate layouts represent 12 plates with specific proficiency panel samples on each plate. All 12 plates were run at least twice for a total of 24 plates for each of Labs C and D.

The analysis of the proficiency panel in the ELISA was performed at Lab E by two separate operators over four days. Samples were analyzed using a starting dilution of 1:50 and followed the plate layout as illustrated in Table 3. This plate layout represents six plates with specific proficiency panel samples on each plate. The six plates were each run four times for a total of 24 plates. For all laboratories, some samples were analyzed three times on the same plate [denoted with "X (3)" in the plate layouts]. These contributed to assay repeatability.

Samples on a given plate were excluded from analysis if the within-assay CV of at least three dilution-adjusted concentrations determined for that sample was greater than 20%. Samples were also excluded if the plate including that sample failed to meet system suitability criteria. Some samples and plates that failed to meet the sample suitability criteria or system suitability criteria were repeated on later days. The ELISA concentrations of each qualification test sample by laboratory are provided in the supplemental information (S1–S6 Tables).

**Table 2. Plate layout used at Laboratories C and D.**

| Sample ID | Plate Number | | | | | |
|---|---|---|---|---|---|---|
| | **1** | **2** | **3** | **4** | **5** | **6** |
| BMI-ZPP-11 | X (3) | X | | X | | X |
| BMI-ZPP-12 | X | X | X | X | X | X |
| BMI-ZPP-13 | | X (3) | | X | X | X |
| BMI-ZPP-14 | X | X | X | X | X | X |
| BMI-ZPP-15 | | X | X (3) | | X | X |
| BMI-ZPP-16 | X | X | X | X | X | X |
| BMI-ZPP-17 | X | | X | X (3) | | X |
| BMI-ZPP-18 | X | X | X | X | X | X |
| BMI-ZPP-19 | X | | X | | X (3) | X |
| BMI-ZPP-20 | X | X | X | X | X | X |

| Sample ID | Plate Number | | | | | |
|---|---|---|---|---|---|---|
| | **7** | **8** | **9** | **10** | **11** | **12** |
| BMI-ZPP-11 | X | X | X | X | X | X |
| BMI-ZPP-12 | X (3) | | | X | X | X |
| BMI-ZPP-13 | X | X | X | X | X | X |
| BMI-ZPP-14 | | X (3) | X | | X | X |
| BMI-ZPP-15 | X | X | X | X | X | X |
| BMI-ZPP-16 | | X | X (3) | X | | X |
| BMI-ZPP-17 | X | X | X | X | X | X |
| BMI-ZPP-18 | X | | X | X (3) | | X |
| BMI-ZPP-19 | X | X | X | X | X | X |
| BMI-ZPP-20 | X | X | | | X (3) | X |

An "X" indicates that sample was analyzed on the indicated plate.

An "X (3)" (shaded) indicates that sample was analyzed on the indicated plate three times.

**Table 3. Plate layout used at Laboratory E.**

| Sample ID | Plate Number | | | | | |
|---|---|---|---|---|---|---|
| | **1** | **2** | **3** | **4** | **5** | **6** |
| BMI-ZPP-11 | X | X | X | X | X | X |
| BMI-ZPP-12 | X (3) | | | X | X | X |
| BMI-ZPP-13 | X | X | X | X | X | X |
| BMI-ZPP-14 | | X (3) | X | | X | X |
| BMI-ZPP-15 | X | X | X | X | X | X |
| BMI-ZPP-16 | | X | X (3) | X | | X |
| BMI-ZPP-17 | X | X | X | X | X | X |
| BMI-ZPP-18 | X | | X | X (3) | | X |
| BMI-ZPP-19 | X | X | X | X | X | X |
| BMI-ZPP-20 | X | X | | | X (3) | X |

An "X" indicates that sample was analyzed on the indicated plate.

An "X (3)" (shaded) indicates that sample was analyzed on the indicated plate three times.

This study, and specifically the use of human serum samples, was approved in writing by the Battelle Institutional Review Board in April of 2015 (approval number HSRE 0223–100062052). Human serum samples were collected from subjects by the sponsor (Crucell Holland) via written consent according to their IRB-approved protocol. These samples were not specifically collected for this interlaboratory study but rather for a different study. Battelle nor any authors were affiliated with this initial study. The sponsor subsequently provided Battelle volumes of these samples for the purposes of conducting the study described in this manuscript. Throughout its analysis of human biological materials and reporting, Battelle had no access to volunteer subjects' identifiers nor any access to any code-key that would allow Battelle researchers to attribute any results of analysis to the original volunteer human research subjects.

## Statistical methods

Inter-laboratory analysis was performed using the combined results across all laboratories. A mixed-effects analysis of variance (ANOVA) model was fitted to the base-10 log-transformed concentrations to evaluate both inter-laboratory precision (i.e., between lab precision) and intra-laboratory precision (i.e., within-laboratory precision). The model included a fixed effect for test sample and random effects for laboratory, test date nested within laboratory, and plate nested within day. Here, test operator was excluded as a random effect because this variable was indistinguishable from test day in most laboratories. Because of this confounding of effects, any variability attributable to test day may also be due to the different test operators.

Results were screened for outliers within each laboratory separately. Deleted studentized residuals were computed for each observation. If the absolute value of the deleted studentized residual was greater than four, then the observation was considered a statistical outlier and removed from the inter-laboratory analysis.

Variability associated with the random effects as well as intermediate precision, repeatability, and total assay variability were estimated separately for each lab using model-based percent coefficient of variation (CV). The percent CV for each source of variance was calculated using Tan's [10] relative standard deviation as

$$100 \times \sqrt{e^{\ln(10)^2 \times \sigma^2} - 1}$$

where $\sigma^2$ is the model-estimated variance for the specific variance source. The percent CV associated with the residual variance served as an estimate for the assay repeatability. The percent CV associated with the test day and plate effects served as an estimate for the intermediate precision of the assay. Total assay variability was estimated using all variance components from the model (both inter- and intra-run variability).

The model intercept was obtained for each test sample from the mixed effects ANOVA model to serve as test sample consensus values across the laboratories. Agreement among laboratories was evaluated by comparing individual assay results from each laboratory to the consensus values. Boxplots were produced for each test sample to show the distribution of concentrations by laboratory in relation to the corresponding consensus value. The ratio of individual test results to consensus values was calculated by test sample to evaluate the level of agreement among laboratories based on two one-sided tests (TOST) of equivalence.

To assess dilutional linearity, a random coefficients linear regression model was fitted to the log-transformed observed concentrations versus the log-transformed target concentrations. The model included both a random intercept and slope effect for each laboratory, along with random effects for laboratory, test day nested within laboratory, and plate nested within laboratory. The random slope coefficients were modeled as laboratory-specific differences

from the overall slope. The overall slope was used to assess the dilutional linearity based on a test of equivalence (TOST) and random slope coefficients were used to evaluate the level of agreement among the laboratories.

## Results

Across all six laboratory runs, there were some false positive observations for Sample 18, a sample with a known negative concentration. All reportable values from Sample 18 were excluded from the statistical models. Table 4 lists five outliers that were removed from their respective intra-laboratory analyses that were also removed from this inter-laboratory analysis. One outlier each were removed from Laboratories B1 and B2. Three outliers were removed from Laboratory C. In the final analysis, Lab A contributed 204 reportable values, Lab B1 had 179 reportable values, Lab B2 had 214 reportable values, Lab C had 268 reportable values, Lab D had 216 reportable values, and Lab E had 218 reportable values.

Table 5 presents ANOVA variance estimates and %CV for each source of variability, intermediate precision, and total assay variability by laboratory. For Laboratory A, the %CV for test date and plate nested within test date were 0.0 and 9.8, respectively. For Laboratory B1, the %CV for test date and plate nested within test date were 10.8 and 15.3, respectively. For Laboratory B2, the %CV for test date and plate nested within test date were 4.5 and 8.9, respectively. For Laboratory C, the %CV for test date and plate nested within test date were 9.8 and 8.5, respectively. For Laboratory D, the %CV for test date and plate nested within test date were 18.9 and 10.5, respectively. Finally, for Laboratory E, the %CVs for test date and plate nested within test date were 7.3 and 5.0, respectively. Laboratory E had the lowest %CV for intermediate precision (8.9) while Laboratory A had the lowest %CV for repeatability (7.2) and total assay variability (12.2). Laboratory B1 had the highest repeatability and total assay variability (23.7%CV and 30.6%CV, respectively) while Laboratory D had the highest %CV for intermediate precision (21.7).

Table 6 shows the consensus values (geometric means) along with 95% confidence intervals for each test sample generated from the mixed model ANOVA fitted to the data. Boxplots by sample of the reportable values from each laboratory, with each plot including a horizontal line for the consensus value estimate for the given sample, are provided in the supplemental information (S1–S9 Figs).

Table 7 shows the ratio of the mean concentration for each of the six individual laboratory runs to the consensus value for a given sample along with a 90% confidence interval for the ratio. Agreement among laboratories implies that these ratios should be close to one, indicating that the average concentrations are about the same as the consensus values. The ratios range from 0.95 to 1.08 for Laboratory A; from 0.96 to 1.19 for Laboratory B1; from 0.83 to 1.12 for Laboratory B2; from 0.96 to 1.16 for Laboratory C; from 0.71 to 0.97 for Laboratory D;

**Table 4.  Statistical outliers identified during analysis of intra-laboratory data.**

| Laboratory | Test Sample | Observed Concentration (ELISA Units/mL) | Target Concentration (ELISA Units/mL) | Studentized Residual |
|---|---|---|---|---|
| B1 | BMI-ZPP-17 | 4.28 | 200 | -9.48 |
| C | BMI-ZPP-13 | 896.47 | 300 | 5.34 |
| C | BMI-ZPP-16 | 236.02 | 500 | -4.74 |
| B2 | BMI-ZPP-19 | 51.20 | 100 | -4.39 |
| C | BMI-ZPP-14 | 1845.88 | 700 | 4.33 |

These observations were deleted from both intra- and inter-laboratory analyses.

**Table 5. Summary of variance components obtained from mixed ANOVA model fit to data from all laboratories (results shown by laboratory).**

| Laboratory A | | |
|---|---|---|
| **Source of Variability** | **Variance** | **%CV** |
| Test Date | 0.0000 | 0.0 |
| Plate Nested in Test Date | 0.0018 | 9.8 |
| Intermediate Precision[1] | 0.0018 | 9.8 |
| Residual (Repeatability) | 0.0010 | 7.2 |
| Total Assay Variability[2] | 0.0028 | 12.2 |

| Laboratory B1 | | |
|---|---|---|
| **Source of Variability** | **Variance** | **%CV** |
| Test Date | 0.0022 | 10.8 |
| Plate Nested in Test Date | 0.0044 | 15.3 |
| Intermediate Precision[1] | 0.0065 | 18.8 |
| Residual (Repeatability) | 0.0103 | 23.7 |
| Total Assay Variability[2] | 0.0169 | 30.6 |

| Laboratory B2 | | |
|---|---|---|
| **Source of Variability** | **Variance** | **%CV** |
| Test Date | 0.0004 | 4.5 |
| Plate Nested in Test Date | 0.0015 | 8.9 |
| Intermediate Precision[1] | 0.0019 | 9.9 |
| Residual (Repeatability) | 0.0033 | 13.3 |
| Total Assay Variability[2] | 0.0052 | 16.7 |

| Laboratory C | | |
|---|---|---|
| **Source of Variability** | **Variance** | **%CV** |
| Test Date | 0.0018 | 9.8 |
| Plate Nested in Test Date | 0.0014 | 8.5 |
| Intermediate Precision[1] | 0.0031 | 13.0 |
| Residual (Repeatability) | 0.0027 | 11.9 |
| Total Assay Variability[2] | 0.0058 | 17.7 |

| Laboratory D | | |
|---|---|---|
| **Source of Variability** | **Variance** | **%CV** |
| Test Date | 0.0066 | 18.9 |
| Plate Nested in Test Date | 0.0021 | 10.5 |
| Intermediate Precision[1] | 0.0087 | 21.7 |
| Residual (Repeatability) | 0.0023 | 11.2 |
| Total Assay Variability[2] | 0.0110 | 24.6 |

| Laboratory E | | |
|---|---|---|
| **Source of Variability** | **Variance** | **%CV** |
| Test Date | 0.0010 | 7.3 |
| Plate Nested in Test Date | 0.0005 | 5.0 |
| Intermediate Precision[1] | 0.0015 | 8.9 |
| Residual (Repeatability) | 0.0015 | 9.0 |
| Total Assay Variability[2] | 0.0030 | 12.7 |

[1]. Comprised of test date and plate nested within test date sources of variability.

[2]. Comprised of repeatability and intermediate precision.

**Table 6. Consensus values by test sample generated from intercept of mixed ANOVA model fit to data from all laboratories.**

| Sample ID | Target Concentration | Consensus Value | 95% CI Consensus Value |
|---|---|---|---|
| BMI-ZPP-11 | 600 | 695.93 | (677.31, 715.06) |
| BMI-ZPP-12 | 400 | 475.20 | (462.47, 488.27) |
| BMI-ZPP-13 | 300 | 325.47 | (316.72, 334.46) |
| BMI-ZPP-14 | 700 | 844.08 | (821.24, 867.55) |
| BMI-ZPP-15 | 800 | 871.34 | (847.91, 895.41) |
| BMI-ZPP-16 | 500 | 561.81 | (546.71, 577.33) |
| BMI-ZPP-17 | 200 | 226.25 | (220.18, 232.49) |
| BMI-ZPP-19 | 100 | 110.63 | (107.66, 113.68) |
| BMI-ZPP-20 | 50 | 70.81 | (68.89, 72.79) |

and from 0.90 to 1.06 for Laboratory E. Fig 1 shows a graph of the mean ratio and 90% confidence interval for each test sample by laboratory.

An equivalence test was conducted to determine if the mean test sample concentrations for each laboratory were equivalent to the corresponding test sample consensus value. An equivalence interval of 0.80 to 1.25 (representing a difference of 20% on the log scale) for the ratio of laboratory mean concentration to consensus concentration was used. The mean laboratory concentration for a given test sample is said to be equivalent to the consensus value for that sample if the 90% confidence interval for the ratio of these two values falls completely within the interval (0.80, 1.25).

**Table 7. Ratio of laboratory mean concentration to overall consensus value with 90% confidence intervals for each test sample.**

| Sample ID | Laboratory A | | Laboratory B1 | | Laboratory B2 | |
|---|---|---|---|---|---|---|
| | Ratio | 90% Confidence Interval | Ratio | 90% Confidence Interval | Ratio | 90% Confidence Interval |
| BMI-ZPP-11 | 1.08 | (1.04, 1.13) | 1.09 | (0.97, 1.23) | 0.83 | (0.81, 0.85) |
| BMI-ZPP-12 | 1.02 | (0.98, 1.08) | 0.97 | (0.87, 1.09) | 1.12 | (1.08, 1.16) |
| BMI-ZPP-13 | 1.02 | (0.98, 1.06) | 1.15 | (1.02, 1.30)* | 0.93 | (0.89, 0.97) |
| BMI-ZPP-14 | 0.99 | (0.96, 1.02) | 0.96 | (0.87, 1.05) | 1.09 | (1.04, 1.13) |
| BMI-ZPP-15 | 1.01 | (0.97, 1.06) | 1.08 | (0.98, 1.18) | 0.88 | (0.84, 0.91) |
| BMI-ZPP-16 | 1.00 | (0.96, 1.04) | 1.07 | (1.00, 1.14) | 1.05 | (1.00, 1.09) |
| BMI-ZPP-17 | 1.06 | (1.01, 1.11) | 1.01 | (0.88, 1.15) | 1.12 | (1.06, 1.18) |
| BMI-ZPP-19 | 0.95 | (0.91, 0.98) | 0.98 | (0.87, 1.09) | 0.83 | (0.79, 0.87)* |
| BMI-ZPP-20 | 0.98 | (0.95, 1.02) | 1.19 | (1.02, 1.39)* | 0.92 | (0.88, 0.97) |
| **Sample ID** | **Laboratory C** | | **Laboratory D** | | **Laboratory E** | |
| | Ratio | 90% Confidence Interval | Ratio | 90% Confidence Interval | Ratio | 90% Confidence Interval |
| BMI-ZPP-11 | 1.10 | (1.05, 1.15) | 0.85 | (0.81, 0.88) | 0.90 | (0.87, 0.94) |
| BMI-ZPP-12 | 1.08 | (1.03, 1.12) | 0.74 | (0.72, 0.77)* | 0.96 | (0.92, 0.99) |
| BMI-ZPP-13 | 1.09 | (1.04, 1.14) | 0.79 | (0.75, 0.83)* | 0.95 | (0.92, 0.99) |
| BMI-ZPP-14 | 1.00 | (0.96, 1.05) | 0.75 | (0.71, 0.80)* | 1.06 | (1.02, 1.10) |
| BMI-ZPP-15 | 1.10 | (1.04, 1.15) | 0.90 | (0.84, 0.97) | 0.98 | (0.95, 1.01) |
| BMI-ZPP-16 | 1.06 | (1.02, 1.11) | 0.76 | (0.73, 0.79)* | 1.01 | (0.97, 1.06) |
| BMI-ZPP-17 | 0.96 | (0.93, 1.00) | 0.71 | (0.69, 0.74)* | 1.02 | (0.99, 1.05) |
| BMI-ZPP-19 | 1.16 | (1.10, 1.22) | 0.97 | (0.94, 1.01) | 0.92 | (0.89, 0.96) |
| BMI-ZPP-20 | 1.10 | (1.02, 1.19) | 0.77 | (0.73, 0.82)* | 1.02 | (0.98, 1.06) |

* 90% confidence interval is outside the acceptance bounds of (0.80, 1.25). Therefore, the concentrations for this test sample are not equivalent to those of other laboratories.

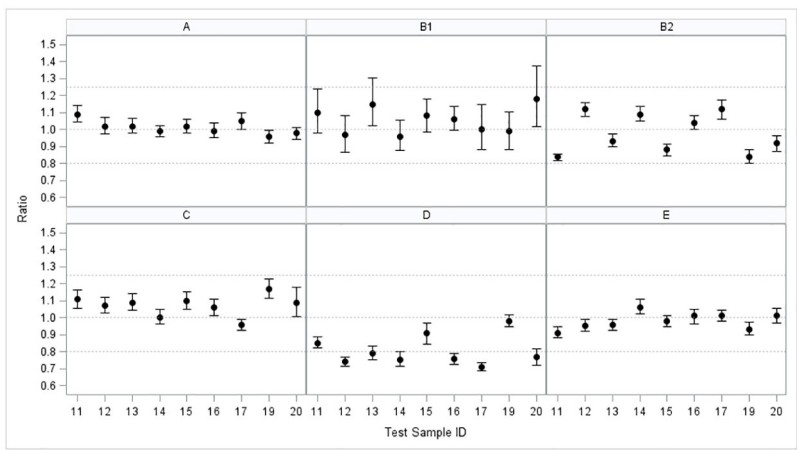

**Fig 1. Graph of ratio of laboratory mean concentration to consensus value with 90% confidence intervals for each test sample by laboratory.** Dotted lines show equivalence region (0.80 to 1.25) and perfect agreement with consensus value (1.00). All means and confidence bounds are entirely within equivalence region for Laboratories A, B2, C, and E.

Following this equivalence criteria: two intervals from Laboratory B1 (corresponding to BMI-ZPP-13 and BMI-ZPP-20) had an upper bound greater than the upper acceptance limit of 1.25 (1.30 and 1.39); one interval from Laboratory B2 (corresponding to BMI-ZPP-19) had a lower bound less than the lower acceptance limit of 0.80 (0.79); and six intervals from Laboratory D (corresponding to BMI-ZPP-12, BMI-ZPP-13, BMI-ZPP-14, BMI-ZPP-16, BMI-ZPP-17, and BMI-ZPP-20) had a lower bound less than the lower acceptance limit of 0.80. Furthermore, three of the six intervals are entirely below the lower acceptance bound of 0.80. These findings indicate that mean concentrations observed at Laboratory D are not equivalent to the other laboratories for six of the nine test samples.

Table 8 presents the estimated slope across the five laboratories and the corresponding 90% confidence interval obtained from the random regression model fit to assess the relationship between $\log_{10}$(observed concentration) and $\log_{10}$(target concentration). The overall slope was estimated to be 0.95 with a 90% confidence interval of (0.93, 0.97). An equivalence test was conducted to determine if the overall slope was equivalent to 1.00 (perfect dilutional linearity). An equivalence interval of 0.80 to 1.25 for the overall slope was used. Because the 90% confidence interval for the overall slope was completely within the interval (0.80, 1.25), the concentrations were found to be dilutionally linear across the laboratories. The slope estimates

**Table 8. Estimated slope and lower and upper 90% confidence interval bounds by laboratory from random coefficients regression model fit to all data.**

| Laboratory | Slope Estimate | 90% Confidence Interval[#] |
|---|---|---|
| Overall (All Labs) | 0.95 | (0.93, 0.97) |
| A | 0.96 | (0.90, 1.02) |
| B1 | 0.94 | (0.88, 1.01) |
| B2 | 0.96 | (0.90, 1.02) |
| C | 0.96 | (0.90, 1.02) |
| D | 0.95 | (0.89, 1.00) |
| E | 0.95 | (0.90, 1.01) |

[#] 90% confidence interval is within the acceptance bounds of (0.80, 1.25). Therefore, the concentrations were dilutionally linear across the laboratories.

specific to each laboratory ranged from 0.94 to 0.96 (Table 8) and were consistent with the overall slope.

## Discussion

The value of an assay as a regulatory tool is dependent on its accuracy, consistency, simplicity, and reproducibility. An assay that is relevant, is species independent, and replicable among laboratories is a powerful tool for product development. The data from a number of clinical trials utilizing ERVEBO strongly suggest that the anti-EBOV GP IgG ELISA provides data that correlate with product efficacy against Ebola infection. The development of new vaccines, or the evaluation of durability or alternative dosing regimens will be based on interpretation of data using the human anti-EBOV GP IgG ELISA. Our ability to use, or trust the data generated from non-clinical studies in different laboratories and clinical trials carried out with sera evaluated at different sites will require an understanding regarding the consistency and reproducibility of the assay among laboratories. In particular, assays using material from animal studies may be performed in laboratories different from those where the assay was performed to evaluate clinical trials. If the assay performance is not consistent among species and across laboratories, then data interpretation will not be possible. This interlaboratory study provided a direct head-to-head comparison of the ELISA performance in five different laboratories. The results from this study confirm the assay can be a universal tool for Ebola virus vaccine evaluation since results were similar when using the assay at multiple labs. However, the small differences in assay performance reinforce that for regulatory purposes, it is still ideal to rely on only one test site where the assay is fully validated.

Intermediate precision for the six laboratory runs ranged from 8.9 to 21.7%CV and repeatability ranged from 7.2 to 23.7%CV. The total assay variability %CVs range from 12.2 to 30.6. As a point of reference, laboratories that validated the anti-EBOV GP IgG ELISA have used the following precision acceptance criteria: (1) The intermediate precision of the assay for samples within the analytic range of the assay must be no larger than 25% CV; and (2) the repeatability of the assay for samples within the analytic range of the assay must be no larger than 20% CV. The repeatability estimate for Laboratory B1 was greater than the upper acceptance bound as established in laboratories that validated the anti-EBOV GP IgG ELISA with human serum. However, a repeat of the proficiency panel run at this laboratory following additional training of laboratory staff resulted in a repeatability estimate less than the upper acceptance bound; thus, illustrating the importance of rigorous training of laboratory staff and the strict adherence to assay procedures to ensure consistent results between runs.

Similarly, laboratories that validated the anti-EBOV GP IgG ELISA have used the following dilutional linearity (relative accuracy) acceptance criteria: the 90% confidence interval for the slope from the random regression model fit to data between the limits of quantitation and relating $\log_{10}$(concentration) to $\log_{10}$(spike level) will be entirely within (-1.20, -0.80). The interlaboratory study models dilutional linearity as $\log_{10}$(observed concentration) to $\log_{10}$(target concentration) resulting in a positive relationship between the two variables. Therefore, to conclude that dilutional linearity is acceptable in relation to the validation in human serum, the 90% confidence interval for the slope should be positive and fall entirely between 0.80 and 1.20. The overall slope was 0.95 and has a 90% confidence interval estimate of (0.93, 0.97); thus, the dilutional linearity is within the acceptance criteria as established in the assay validation with human serum.

Agreement among laboratories implies that the ratios of the mean concentration for the five individual labs to the overall laboratory consensus value for a given sample should be close to one. The ratios range from 0.95 to 1.08 for Laboratory A; from 0.96 to 1.19 for Laboratory

B1; from 0.83 to 1.12 for Laboratory B2; from 0.96 to 1.16 for Laboratory C; from 0.71 to 0.97 for Laboratory D; and from 0.90 to 1.06 for Laboratory E. Equivalence test results showed that the 90% confidence interval for the ratio were within the equivalence bounds of 0.80 to 1.25 for each laboratory except for samples BMI-ZPP-13 and BMI-ZPP-20 in Laboratory B1, BMI-ZPP-19 in Laboratory B2, and six samples in Laboratory D.

The assessment of between-laboratory performance revealed lower observed concentrations at Lab D and greater variability in assay results at Lab B1 relative to the other laboratories. The lower observed concentrations at Lab D illustrate the importance of monitoring assay performance and harmonizing across laboratories. Given the inherent differences from subject-to-subject in clinical trials and animal-to-animal in non-clinical studies, these differences observed at Lab D relative to the other laboratories are not likely to affect interpretation of study results. The variability in assay results at Lab B1 was mitigated by additional laboratory staff training.

The evaluation of the proficiency panel at these laboratories provides a limited assessment of assay precision (intermediate precision, repeatability, and total assay variability), dilutional linearity, and accuracy. This limited evaluation suggests that the within-laboratory performance of anti-EBOV GP IgG ELISA as implemented at the five laboratories is performing consistently with the intended use of the assay based on the acceptance criteria used by laboratories that have validated the assay.

## Supporting information

**S1 Fig. Observed concentration (ELISA Units/mL) by laboratory for sample BMI-ZPP-11.** (Consensus Concentration = 695.93). Center line in the box depicts the median concentration while the height of the box represents the 25th and 75th percentile of the concentration distribution. Vertical lines extending above and below the box represent the maximum and minimum concentration values for the laboratory. Open circles show the observed concentrations. (TIF)

**S2 Fig. Observed concentration (ELISA Units/mL) by laboratory for sample BMI-ZPP-12.** (Consensus Concentration = 475.20). Center line in the box depicts the median concentration while the height of the box represents the 25th and 75th percentile of the concentration distribution. Vertical lines extending above and below the box represent the maximum and minimum concentration values for the laboratory. Open circles show the observed concentrations. (TIF)

**S3 Fig. Observed concentration (ELISA Units/mL) by laboratory for sample BMI-ZPP-13.** (Consensus Concentration = 325.47). Center line in the box depicts the median concentration while the height of the box represents the 25th and 75th percentile of the concentration distribution. Vertical lines extending above and below the box represent the maximum and minimum concentration values for the laboratory. Open circles show the observed concentrations. (TIF)

**S4 Fig. Observed concentration (ELISA Units/mL) by laboratory for sample BMI-ZPP-14.** (Consensus Concentration = 844.08). Center line in the box depicts the median concentration while the height of the box represents the 25th and 75th percentile of the concentration distribution. Vertical lines extending above and below the box represent the maximum and minimum concentration values for the laboratory. Open circles show the observed concentrations. (TIF)

**S5 Fig. Observed concentration (ELISA Units/mL) by laboratory for sample BMI-ZPP-15.** (Consensus Concentration = 871.34). Center line in the box depicts the median concentration while the height of the box represents the 25th and 75th percentile of the concentration distribution. Vertical lines extending above and below the box represent the maximum and minimum concentration values for the laboratory. Open circles show the observed concentrations. (TIF)

**S6 Fig. Observed concentration (ELISA Units/mL) by laboratory for sample BMI-ZPP-16.** (Consensus Concentration = 561.81). Center line in the box depicts the median concentration while the height of the box represents the 25th and 75th percentile of the concentration distribution. Vertical lines extending above and below the box represent the maximum and minimum concentration values for the laboratory. Open circles show the observed concentrations. (TIF)

**S7 Fig. Observed concentration (ELISA Units/mL) by laboratory for sample BMI-ZPP-17.** (Consensus Concentration = 226.25). Center line in the box depicts the median concentration while the height of the box represents the 25th and 75th percentile of the concentration distribution. Vertical lines extending above and below the box represent the maximum and minimum concentration values for the laboratory. Open circles show the observed concentrations. (TIF)

**S8 Fig. Observed concentration (ELISA Units/mL) by laboratory for sample BMI-ZPP-19.** (Consensus Concentration = 110.63). Center line in the box depicts the median concentration while the height of the box represents the 25th and 75th percentile of the concentration distribution. Vertical lines extending above and below the box represent the maximum and minimum concentration values for the laboratory. Open circles show the observed concentrations. (TIF)

**S9 Fig. Observed concentration (ELISA Units/mL) by laboratory for sample BMI-ZPP-20.** (Consensus Concentration = 70.81). Center line in the box depicts the median concentration while the height of the box represents the 25th and 75th percentile of the concentration distribution. Vertical lines extending above and below the box represent the maximum and minimum concentration values for the laboratory. Open circles show the observed concentration. (TIF)

**S1 Table. ELISA concentration of each test sample—Laboratory A.** (XLSX)

**S2 Table. ELISA concentration of each test sample—Laboratory B1.** (XLSX)

**S3 Table. ELISA concentration of each test sample—Laboratory B2.** (XLSX)

**S4 Table. ELISA concentration of each test sample—Laboratory C.** (XLSX)

**S5 Table. ELISA concentration of each test sample—Laboratory D.** (XLSX)

**S6 Table. ELISA concentration of each test sample—Laboratory E.** (XLSX)

## Acknowledgments

Opinions, interpretations, conclusions, and recommendations are those of the authors and are not necessarily endorsed by the U.S. Army. We acknowledge the members of the FANG EBOV ELISA community and the participating laboratories for their insightful questions, comments, and recommendations during the performance of this interlaboratory evaluation of the assay.

## Author Contributions

**Conceptualization:** Michael S. Anderson, Carol L. Sabourin, Thomas L. Rudge, Jr.

**Data curation:** Michael S. Anderson, Nancy A. Niemuth, Thomas L. Rudge, Jr.

**Formal analysis:** Michael S. Anderson, Nancy A. Niemuth.

**Investigation:** Michael S. Anderson, Carol L. Sabourin, Thomas L. Rudge, Jr.

**Methodology:** Michael S. Anderson, Nancy A. Niemuth, Carol L. Sabourin, Thomas L. Rudge, Jr.

**Project administration:** Michael S. Anderson, Carol L. Sabourin, Thomas L. Rudge, Jr.

**Resources:** Callie E. Bounds.

**Supervision:** Michael S. Anderson.

**Validation:** Michael S. Anderson.

**Writing – original draft:** Michael S. Anderson, Christopher S. Badorrek, Callie E. Bounds, Thomas L. Rudge, Jr.

**Writing – review & editing:** Michael S. Anderson, Nancy A. Niemuth, Carol L. Sabourin, Christopher S. Badorrek, Callie E. Bounds, Thomas L. Rudge, Jr.

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
