## [Decision Letter · Decision Letter 0]

3 Jun 2020

PONE-D-20-10369

Interlaboratory comparison for the Filovirus Animal Nonclinical Group (FANG) anti-Ebola virus glycoprotein immunoglobulin G enzyme-linked immunosorbent assay

PLOS ONE

Dear Dr. Anderson,

Thank you for submitting your manuscript to PLOS ONE. After careful consideration, we feel that it has merit but does not fully meet PLOS ONE’s publication criteria as it currently stands. Therefore, we invite you to submit a revised version of the manuscript that addresses the points raised during the review process.

We look forward to receiving your revised manuscript.

Kind regards,

John Schieffelin, MD

Academic Editor

PLOS ONE

Journal Requirements:

2. Please provide additional details regarding human serum samples used in your study. In the ethics statement in the Methods and online submission information, please ensure that you have specified (1) the sources, e.g. are they collected from human subjects specifically recruited for your study? (2) if so, whether consent was informed and what type you obtained (for instance, written or verbal, and if verbal, how it was documented and witnessed). If the need for consent was waived by the ethics committee, please include this information.”

"The authors have declared that no competing interests exist.".

We note that one or more of the authors are employed by a commercial company: 'Battelle'

Reviewers' comments:

Reviewer's Responses to Questions

**Comments to the Author**

1. Is the manuscript technically sound, and do the data support the conclusions?

Reviewer #1: Yes

Reviewer #2: Yes

2. Has the statistical analysis been performed appropriately and rigorously? 

Reviewer #1: Yes

Reviewer #2: Yes

3. Have the authors made all data underlying the findings in their manuscript fully available?

Reviewer #1: Yes

Reviewer #2: Yes

4. Is the manuscript presented in an intelligible fashion and written in standard English?

Reviewer #1: Yes

Reviewer #2: Yes

5. Review Comments to the Author

Reviewer #1: Line 22: first sentence of the abstract is long, consider breaking up into 2 points about the West African outbreak and then additionally the ongoing outbreak in DRC

Line 27: mention that the anti-EBOV GP IgG ELISA is the FANG assay here

Line 54: again I think at least initially (even though your cite it) you should say that the ELISA is the FANG assay.

Some of the information in the Introduction sounds more like study design which should be part of the Methods. I would like to see more of a background here in the Introduction section perhaps on the development of the assay initially, why this assay was chosen, rationale for doing the study, etc. I think a bit more context for the study is needed in this section.

I would like to see more expansion on the implications of the results in the discussion section. For example spell out the implications of the lower observed concentrations at Lab D and the variability of results in Lab B1. What do these results mean for the use of this assay moving forward?

This manuscript provides important information on assay validation of the FANG assay, an anti-EBOV GP IgG ELISA. These studies are needed in order to ensure robust assays and provide confidence in interpreting data between and within laboratories. My main concern with this manuscript is that more information is needed in both the Introduction and the Discussion to better understand the context of this assay and why the study is warranted. I think it is very much warranted but would like to see more of this information presented in the text.

Reviewer #2: Anderson et al provide data examining the performance of the FANG IgG EBOV ELISA across multiple sites. The study was intended to assess inter and intra-laboratory performance of the assay. The resulting data suggest that the assay is relatively robust, but does highlight a few limitations that the authors claim tend to be laboratory specific. The study has value for assessing preclinical/clinical samples for vaccine development across multiple institutions and clinical sites; where uniformity and agreement among analytical methods are critical for evaluating performance of a given countermeasure, in this case, vaccines targeting the EBOV GP. The paper was digestible and fairly straight forward.

A point of clarification:

Abstract, line 26: One vaccine is fully licensed and not considered investigational, the other is investigational at present.

Please go through the article and verify issues with spelling and grammar throughout to ensure clean readability.

6. PLOS authors have the option to publish the peer review history of their article (what does this mean?). If published, this will include your full peer review and any attached files.

Reviewer #1: No

Reviewer #2: No

---

## [Author Response · Author response to Decision Letter 0]

15 Jul 2020

1. Comment: Please ensure that your manuscript meets PLOS ONE's style requirements, including those for file naming.

Response: We have verified that the manuscript adheres to PLOS ONE’s style requirements.

2. Comment: Please provide additional details regarding human serum samples used in your study. In the ethics statement in the Methods and online submission information, please ensure that you have specified (1) the sources, e.g. are they collected from human subjects specifically recruited for your study? (2) if so, whether consent was informed and what type you obtained (for instance, written or verbal, and if verbal, how it was documented and witnessed). If the need for consent was waived by the ethics committee, please include this information.”

Response: Human serum samples were collected from subjects by the sponsor (Crucell Holland) via written consent according to their IRB-approved protocol. These samples were not specifically collected for this interlaboratory study but rather for a different study (“A Phase 1, Randomized, Placebo-Controlled, Observer-Blind Study to Evaluate the Safety, Tolerability and Immunogenicity of Heterologous Prime-Boost Regimens Using MVA-BN®-Filo and Ad26.ZEBOV Administered in Different Doses, Sequences, and Schedules in Healthy Adult Subjects”). Battelle was not involved in this study. The sponsor subsequently provided Battelle volumes of these samples for the purposes of conducting the study described in this manuscript. Throughout its analysis of human biological materials and reporting, Battelle had no access to volunteer subjects’ identifiers nor any access to any code-key that would allow Battelle researchers to attribute any results of analysis to the original volunteer human research subjects. The ethics statement in the Methods Section and online submission form has been updated with the requested information.

3. Comment: Please provide an amended Funding Statement declaring this commercial affiliation, as well as a statement regarding the Role of Funders in your study. If the funding organization did not play a role in the study design, data collection and analysis, decision to publish, or preparation of the manuscript and only provided financial support in the form of authors' salaries and/or research materials, please review your statements relating to the author contributions, and ensure you have specifically and accurately indicated the role(s) that these authors had in your study. You can update author roles in the Author Contributions section of the online submission form.

Response: We have provided the following amended Funding Statement in our cover letter: 

“This project has been funded in whole or in part with funds from the U.S. Department of Defense (DOD) Joint Program Executive Office for Chemical, Biological, Radiological, and Nuclear Defense (JPEO-CBRND) Medical Countermeasure Systems Joint Vaccine Acquisition Program (MCS-JVAP) under Battelle contract number GS00Q140ADU402, delivery order W911QY-16-F-0074. The funding body provided support in the form of salaries for some of the authors (MSA, NAN, CLS, TLR) and contributed to study design and in the decision to submit the article for publication. Our (MSA, NAN, TLR) affiliation with Battelle, a non-profit contract research organization, did not play a role in the study. The specific roles of these authors are articulated in the ‘author contributions’ section. The funding body contributed to study design and in the decision to submit the article for publication. The views expressed here are those of the authors and do not necessarily represent the views or official position of the DOD.”

4. Comment: Please also include the following statement within your amended Funding Statement: “The funder provided support in the form of salaries for authors [insert relevant initials], but did not have any additional role in the study design, data collection and analysis, decision to publish, or preparation of the manuscript. The specific roles of these authors are articulated in the ‘author contributions’ section.” If your commercial affiliation did play a role in your study, please state and explain this role within your updated Funding Statement.

Response: Our new funding statement contains this language.

5. Comment: Please also provide an updated Competing Interests Statement declaring this commercial affiliation along with any other relevant declarations relating to employment, consultancy, patents, products in development, or marketed products, etc. Within your Competing Interests Statement, please confirm that this commercial affiliation does not alter your adherence to all PLOS ONE policies on sharing data and materials by including the following statement: "This does not alter our adherence to PLOS ONE policies on sharing data and materials.” (as detailed online in our guide for authors http://journals.plos.org/plosone/s/competing-interests) . If this adherence statement is not accurate and there are restrictions on sharing of data and/or materials, please state these. Please note that we cannot proceed with consideration of your article until this information has been declared.

Response: We have provided the following amended Competing Interests Statement in our cover letter:

All authors listed on the manuscript have read the manuscript and approve its submission to PLOS ONE. We have no issues relevant to your journal policies for publication, and the authors have no potential competing interests to declare. Our (MSA, NAN, TLR) employment by Battelle, a non-profit contract research organization, does not alter our adherence to PLOS ONE policies on sharing data and materials nor played a role in the study. This manuscript has not been published previously and is not under consideration for publication in any other scientific journal.

6. Comment: Line 22: first sentence of the abstract is long, consider breaking up into 2 points about the West African outbreak and then additionally the ongoing outbreak in DRC.

Response: The first sentence has been broken into two separate statements.

7. Comment: Line 27: mention that the anti-EBOV GP IgG ELISA is the FANG assay here.

Response: The mention has been added to the manuscript.

8. Comment: Again I think at least initially (even though your cite it) you should say that the ELISA is the FANG assay. Some of the information in the Introduction sounds more like study design which should be part of the Methods. I would like to see more of a background here in the Introduction section perhaps on the development of the assay initially, why this assay was chosen, rationale for doing the study, etc. I think a bit more context for the study is needed in this section.

Response: We added language to clarify that the ELISA is the FANG assay. Text has also been added to the Introduction section to include some background of the assay development.

9. Comment: I would like to see more expansion on the implications of the results in the discussion section. For example spell out the implications of the lower observed concentrations at Lab D and the variability of results in Lab B1. What do these results mean for the use of this assay moving forward?

Response: The requested information has been added to the Discussion section.

10. Comment: This manuscript provides important information on assay validation of the FANG assay, an anti-EBOV GP IgG ELISA. These studies are needed in order to ensure robust assays and provide confidence in interpreting data between and within laboratories. My main concern with this manuscript is that more information is needed in both the Introduction and the Discussion to better understand the context of this assay and why the study is warranted. I think it is very much warranted but would like to see more of this information presented in the text.

Response: The requested additional information has been added to the Introduction and Discussion sections.

11. Comment: Abstract, line 26: One vaccine is fully licensed and not considered investigational, the other is investigational at present.

Response: This clarification has been made in the manuscript.

12. Comment: Please go through the article and verify issues with spelling and grammar throughout to ensure clean readability.

Response: We have gone through and verified there are no issues with spelling and grammar issues that would hinder clean readability of the manuscript.

---

## [Decision Letter · Decision Letter 1]

12 Aug 2020

Interlaboratory comparison for the Filovirus Animal Nonclinical Group (FANG) anti-Ebola virus glycoprotein immunoglobulin G enzyme-linked immunosorbent assay

PONE-D-20-10369R1

Dear Dr. Anderson,

We’re pleased to inform you that your manuscript has been judged scientifically suitable for publication and will be formally accepted for publication once it meets all outstanding technical requirements.

Kind regards,

John Schieffelin, MD

Academic Editor

PLOS ONE

Additional Editor Comments (optional):

Reviewers' comments:

Reviewer's Responses to Questions

**Comments to the Author**

1. If the authors have adequately addressed your comments raised in a previous round of review and you feel that this manuscript is now acceptable for publication, you may indicate that here to bypass the “Comments to the Author” section, enter your conflict of interest statement in the “Confidential to Editor” section, and submit your "Accept" recommendation.

Reviewer #1: All comments have been addressed

2. Is the manuscript technically sound, and do the data support the conclusions?

Reviewer #1: (No Response)

3. Has the statistical analysis been performed appropriately and rigorously? 

Reviewer #1: (No Response)

4. Have the authors made all data underlying the findings in their manuscript fully available?

Reviewer #1: (No Response)

5. Is the manuscript presented in an intelligible fashion and written in standard English?

Reviewer #1: (No Response)

6. Review Comments to the Author

Reviewer #1: (No Response)

7. PLOS authors have the option to publish the peer review history of their article (what does this mean?). If published, this will include your full peer review and any attached files.

Reviewer #1: No

---

## [Editor Report · Acceptance letter]

14 Aug 2020

PONE-D-20-10369R1 

Interlaboratory comparison for the Filovirus Animal Nonclinical Group (FANG) anti-Ebola virus glycoprotein immunoglobulin G enzyme-linked immunosorbent assay 

Dear Dr. Anderson:

I'm pleased to inform you that your manuscript has been deemed suitable for publication in PLOS ONE. Congratulations! Your manuscript is now with our production department. 

Kind regards, 

on behalf of

Dr, John Schieffelin 

Academic Editor

PLOS ONE